# Development of a Multiplex Polymerase Chain Reaction-Based DNA Lateral Flow Assay as a Point-of-Care Diagnostic for Fast and Simultaneous Detection of MRSA and Vancomycin Resistance in Bacteremia

**DOI:** 10.3390/diagnostics12112691

**Published:** 2022-11-04

**Authors:** Mona T. Kashef, Omneya M. Helmy

**Affiliations:** Department of Microbiology and Immunology, Faculty of Pharmacy, Cairo University, Cairo 11562, Egypt

**Keywords:** bacteremia, diagnosis, lateral flow assay, MRSA, multiplex, point-of-care, polymerase chain reaction, *Staphylococcus aureus*, vancomycin, resistance

## Abstract

To reduce high mortality and morbidity rates, timely and proper treatment of methicillin-resistant *Staphylococcus aureus* (MRSA) bloodstream infection is required. A multiplex polymerase reaction (mPCR)-based DNA lateral flow assay (MBDLFA) was developed as a point-of-care diagnostic for simultaneous identification of *S. aureus*, methicillin resistance, and vancomycin resistance directly from blood or blood cultures. A mPCR was developed to detect *nuc*, *mec*A, and *van*A/B; its sensitivity, specificity, and limit of detection (LOD) were determined. The developed reaction was further modified for use in MBDLFA and its sensitivity for detection of target genes from artificially inoculated blood samples was checked. The optimized mPCR successfully detected *nuc*, *mec*A, and *van*A/B from genomic DNA of bacterial colonies with LODs of 10^7^, 10^7^, and 10^5^ CFU/mL, respectively. The reaction was sensitive and specific. The optimized mPCR was used in MBDLFA that detected *nuc*, *mec*A, and *van*A/B with LODs of 10^7^, 10^8^, and 10^4^ CFU/mL, respectively, directly from artificially inoculated blood. The developed MBDLFA can be used as a rapid, cheap point-of-care diagnostic for detecting *S. aureus*, MRSA, and vancomycin resistance directly from blood and blood cultures in ~2 h with the naked eye. This will reduce morbidity, mortality, and treatment cost in *S. aureus* bacteremia.

## 1. Introduction

Bacteremia or bloodstream infection is a term used to define the presence of bacteria in blood with the development of infection signs [1]. It is usually associated with high rates of morbidity and mortality. Bloodstream infection rates were estimated to reach two million episodes yearly in North America and Europe combined, with an estimated one-quarter of a million deaths [2]. *Escherichia coli* has the highest incidence, whereas *Staphylococcus aureus* and *Pseudomonas aeruginosa* cause higher mortality rates [3].

*S. aureus* ranked second among pathogens causing bacteremia; *S. aureus* bacteremia (SAB) is associated with a mortality rate reaching 30% [4]. This high mortality rate usually arises from the serious associated complications such as endocarditis, osteomyelitis, and metastatic abscesses [5]. Several factors affect the mortality rates of bacteremia. Some of these factors are related to the patients themselves, such as age, immune status, ethnicity, and comorbidities. Other factors are related to host–pathogen interaction, such as persistent bacteremia, bacteriuria, or shock. The mortality rate of SAB is also affected by toxin production, *S. aureus* clonal type, and the antimicrobial resistance against methicillin, where methicillin-resistant *S. aureus* (MRSA) is associated with a significant increase in mortality [6,7]. Recently, a higher mortality rate is associated with SAB in patients with COVID-19 infection [8].

This increase in mortality in MRSA bacteremia may arise from increased virulence caused by the presence of SCC*mec*-associated virulence factors [7] or from the delayed receipt of appropriate antimicrobials or improper therapy [9]. Management of SAB depends mainly on the antimicrobial resistance pattern of *S. aureus*, i.e., whether it is methicillin-sensitive *S. aureus* (MSSA) or MRSA. MSSA is better treated with semisynthetic penicillins or first-generation cephalosporins rather than vancomycin. However, vancomycin remains the drug of choice for treating MRSA bacteremia and is used as empirical therapy when bloodstream infection is suspected [10,11]. Treatment failure associated with vancomycin use has also been reported, although not at a high rate [11].

Proper antibiotic administration timing is another factor affecting SAB’s outcome [7]. Using rapid diagnostic testing to identify the causative agents and their susceptibilities has reduced the mortality risk in bloodstream infections [12]. Therefore, proper and timely identification of *S. aureus* and its antimicrobial susceptibility pattern is critical in reducing the mortality and morbidity associated with SAB. Unfortunately, using culture-dependent methods for identification and susceptibility testing is time-consuming and the identification results may not be available for three days. Blood culturing is the gold standard for bacteremia diagnosis; however, the blood culture results may not be available before the next day of testing, considering the working hours for sample processing and reporting [6].

Polymerase chain reaction (PCR)-based tests have provided a rapid means for microbial identification without the need for timely culture methods [13]. However, post-PCR steps, in their simplest form, using gel electrophoresis, are time-consuming and labor-intensive, especially when used for many samples, besides using the carcinogenic ethidium bromide for visualization [14]. Other molecular techniques for microbial identification include pyrosequencing and fluorescent-probe hybridization; however, these methods are expensive, labor-intensive, and require specific instrumentation unavailable in the laboratories of countries with limited resources [13]. Many techniques were developed to simplify the post-PCR steps such as DNA biosensors, where a DNA probe is immobilized onto the sensor’s surface. In these techniques, PCR can be carried out with primers containing specific oligonucleotide sequences or immunogenic substances bound to their 5′ or 3′ ends followed by hybridization to the DNA probe and DNA detection using optical, electrochemical, or gravimetric means [15], where labels such as colored latex particles, gold nanoparticles, or florescent compounds are used for detection [16].

The development of point-of-care diagnostics has gained great attention. They allow for optimized treatment and minimize the cost and time required for sample transport to central laboratories as they need simple instrumentation [17]. The nucleic acid lateral flow assay (LFA) is an example of a DNA biosensor that can be used as a point-of-care diagnostic allowing the rapid and specific DNA identification at the patient’s site. It is user-friendly, operates at a low cost, requires simple instrumentation, and allows for a one-step analysis. Several types and techniques are available for the nucleic acid lateral flow assay. These assays have many applications, such as clinical analysis and pathogen and toxin detection, as well as identification of various proteins, pesticides, and heavy metals in medicine and the environment [16]. Lateral flow assays were developed for microbial identification and detection of their resistance to antimicrobials in stool [18]. They were also used in environmental DNA detection [19].

DNA lateral flow assays depending on single-tag hybridization technology were developed in the form of dipstick strips for the detection of PCR amplicons. Single-tag hybridization depends on using biotinylated oligonucleotides and a single-stranded tag–spacer sequence for preparing the PCR amplicons that bind streptavidin-coated blue latex particles. The product is then hybridized to a complementary probe immobilized on the membrane strip, producing a colored line [20]. The membrane test strip is a nitrocellulose paper with different pads that allow capillary sample flow. The sample is applied onto a sample pad and moves by capillary flow to reach the conjugate pad, where it reacts with the complementary probe, thus allowing product visualization. Using a multiplex format allows the detection of different products on a single strip using different tags, which is advantageous in terms of time and cost savings [16].

Here, we developed a multiplex polymerase reaction (mPCR)-based DNA lateral flow assay (MBDLFA) as a point-of-care diagnostic for the simultaneous identification of *S. aureus*, methicillin resistance, and vancomycin resistance directly from blood samples or blood cultures of suspected bacteremia patients.

## 2. Materials and Methods

### 2.1. Bacterial Strains and Culture Conditions

Clinical isolates of methicillin-resistant *S. aureus* (S31, S43) from the culture collection of the Department of Microbiology and Immunology, Faculty of Pharmacy, Cairo University, and vancomycin-resistant *Enterococcus faecalis* (E25, Y3), kindly provided by Dr. Yomna Hashem, were used in this study. Other standard microbial species were used in the specificity testing: *S. aureus* ATCC 25923, *Acinetobacter baumannii* ATCC 19606, *E. coli* ATCC 25922, *Enterococcus faecium* ATCC 27270, *Enterococcus faecalis* ATCC 19433, *Klebsiella pneumoniae* ATCC 10031, and *P. aeruginosa* ATCC 27856. All microbial strains were stored in Luria Bertani broth containing 25% glycerol at –70 °C; when required, they were retrieved from the frozen stock by subculturing onto Luria Bertani agar plates and incubating at 37 °C for 24 h.

### 2.2. Primer Design

Primer pairs were designed to allow partial amplification of conserved regions in *nuc*, *mec*A, and vancomycin-resistance gene variants A and B (*van*A/B). The *nuc* gene encodes for a thermonuclease specific for *S. aureus* [21]; *mec*A encodes for penicillin-binding protein 2 that is responsible for methicillin resistance [22], and *van*A/B encodes for vancomycin-resistance determinants type A and B [23]. Different sequences of the target genes were downloaded from the National Center for Biotechnology Information (NCBI); the list of accession numbers of the used sequences is provided in Appendix A. The sequences of each gene were aligned and primers were designed in the conserved regions [24,25] (Table 1). The primers were designed so that their parameters and melting temperatures allowed their use in a multiplex amplification reaction with product sizes that differ by at least 100 bp. Primer specificity was confirmed using the Primer-BLAST tool (NCBI, https://www.ncbi.nlm.nih.gov/, last accessed January 2022). All primers used in the uniplex and the conventional multiplex polymerase chain reactions (mPCR) were commercially synthesized by Macrogen, Korea.

### 2.3. Conventional Multiplex PCR Development

The DNA was extracted from the test isolates (S31 and E25) using DNeasy blood and tissue kit (Qiagen, Germany) according to the manufacturer’s instructions. Initially, the designed primers were checked in a uniplex PCR (25 µL) consisting of: 1 µL of extracted DNA, 5 µL of 5X GoTaq Flexi Reaction Buffer, 1.7 mM MgCl_2_, 0.2 mM dNTP mix, 20 μM primer (Table 1), and 0.625 U GoTaq DNA Polymerase. All PCR reagents were from Promega (Madison, WI, USA). The reaction was performed in Veriti 96-Well Fast thermal cycler (Applied Biosystems ^TM^, San Francisco, CA, USA) using initial denaturation at 94 °C for 3 min, then 30 cycles of denaturation at 94 °C for 30 s, annealing at 60 °C for 30 s, and extension at 72 °C for 1 min, followed by a final extension step at 72 °C for 10 min. The amplicons were detected by electrophoresis in ethidium bromide-stained agarose gels (1.5%). The PCR products were purified using Gene JET™ PCR Purification Kit (Thermo Fisher Scientific, Vilnius, Lithuania, EU) and sequenced using an ABI3730XL sequencer (Macrogen, Seoul, Korea). The resulting nucleotide sequences were checked for similarity to confirm their specificity using the BlastN tool available at NCBI (https://blast.ncbi.nlm.nih.gov/Blast.cgi?PROGRAM=tblastn&PAGE_TYPE=BlastSearch&LINK_LOC=blasthome, accessed on August 2022) and were deposited in Genbank.

The primers were then used to develop a conventional mPCR. Optimizing mPCR involved testing different Mg^2+^ concentrations and a gradient of annealing temperatures ranging from 55–60 °C. The optimum mPCR reaction mixture (25 µL) contained 1 µL of extracted DNA, 5 µL of 5X Green GoTaq Flexi Reaction Buffer, 1.7 mM MgCl_2_, 0.2 mM dNTP mix, 20 μM of each primer (Table 1), and 0.625 U GoTaq DNA Polymerase. The cycling parameters were as follows: initial denaturation at 94 °C for 3 min, then 30 cycles of denaturation at 94 °C for 30 s, annealing at 60 °C for 30 s, and extension at 72 °C for 1 min, followed by a final extension step at 72 °C for 10 min. The mPCR amplicons were visualized by electrophoresis on 1.5% agarose gels stained with ethidium bromide.

### 2.4. Sensitivity and Limit of Detection (LOD) of the Developed mPCR

The total aerobic viable counts of overnight cultures of S31 and E25 isolates were determined. The genomic DNA was extracted from different dilutions of S31 and E25 suspensions (from 10^7^ to 10^3^ CFU/mL in phosphate-buffered saline) using the DNeasy Blood and Tissue Kit (Qiagen, Germany), according to manufacturer’s instructions. The optimized mPCR reaction was then performed on the extracted DNA from each dilution, and the PCR products were visualized by electrophoresis using a 1.5% agarose gel stained with ethidium bromide. The LOD of the mPCR was determined as the smallest concentration of cells with detectable products [26].

### 2.5. Specificity of the Developed mPCR

The specificity of the developed mPCR was tested using *S. aureus* ATCC 25923, MRSA strain S43, vancomycin-resistant *E. faecalis* Y3, as well as other bacteremia-causing pathogens (*A. baumannii* ATCC 19606, *E. coli* ATCC 25922, *E. faecium* ATCC 27270, *E. faecalis* ATCC 19433, *K. pneumoniae* ATCC 10031, and *P. aeruginosa* ATCC 27856). Genomic DNA was extracted from the tested strains using the DNeasy Blood and Tissue Kit, according to the manufacturer’s instructions. The optimized mPCR was performed on the extracted DNA as described earlier. Also, the assay was performed on pooled extracted DNA from S31 and E25.

### 2.6. Development of the mPCR-Based DNA Lateral Flow Assay (MBDLFA)

The primers for the MBDLFA were synthesized using the sequences of the previously designed primers for conventional mPCR. The 5′ terminus of the forward primer contained an oligonucleotide sequence complementary to the oligonucleotide immobilized on the corresponding line of the custom-designed chromatography printed array strip (C-PAS), and the 5′ terminus of the reverse primer was biotinylated (Table 1). All primers, buffers, and test strips were commercially synthesized by TBA Co., Japan. Optimizing the mPCR for use in the MBDLFA involved testing different primers and Mg^2+^ concentrations, a gradient of annealing temperatures ranging from 55–60 °C, and high and low concentrations of DNA. The optimized reaction (25 µL) contained 1 µL of extracted DNA, 5 µL of 5X GoTaq Flexi Reaction Buffer, 2 mM MgCl_2_, 0.2 mM dNTP mix, 7.5 μM of each of Tag1-NucF and Biotin-NucR primer, 5 μM of each of Tag2-mecAF and Biotin-mecAR primer, 20 μM each of Tag3-VanF and Biotin-VanR primers, and 0.625 U GoTaq DNA Polymerase and used the same cycling parameters as the conventional mPCR. The mPCR products for the MBDLFA (10 µL) were mixed with 10 µL dilution buffer, and 1 µL of latex solution containing avidin-coated blue beads. According to the manufacturer’s protocol, optimizing the lateral flow dilution buffer involved testing dilution buffers with different NaCl concentrations (Modi; 0 mM NaCl incl, 150 mM NaCl incl, and 300 mM NaCl incl); using Modi; 0 mM NaCl was optimum. The designed C-PAS was immersed in the mixture and the results were visualized after a 5 min incubation period at room temperature via the presence of blue-colored lines corresponding to each tested gene (Figure 1). The MBDLFA was tested on the DNA extracted from bacterial colonies of S31 and E25 as well as on pooled DNA of S31 and E25. The specificity of the developed MBDLFA was confirmed as previously described under conventional mPCR using different Gram-positive and Gram-negative pathogens.

### 2.7. Detection of MRSA and Vancomycin Resistance in Artificially Spiked Blood Samples Using the MBDLFA

A defibrinated human blood sample was artificially inoculated with either S31 or E25 [27]. Briefly, overnight cultures of S31 or E25 were diluted to reach a viable count equivalent to 10^7^ CFU/mL. Ten-fold serial dilutions were made in Luria Bertani broth, 1 mL of different dilutions of the bacterial culture was centrifuged at 13,000 rpm for 5 min, and the pellet was suspended in 100 µL defibrinated blood. DNA was extracted from each inoculated blood sample using the DNeasy Blood and Tissue Kit, according to the manufacturer’s protocol. The developed MBDLFA was performed using 1 µL of each extracted DNA sample, as described previously. The results of the developed MBDLFA were visualized via the naked eye by two independent technicians to avoid any bias during result interpretation. The products were also visualized by electrophoresis using 1.5% agarose gel stained with ethidium bromide.

## 3. Results

### 3.1. Development of mPCR for the Detection of S. aureus, Methicillin Resistance and Vancomycin Resistance

The designed primers were tested in three separate uniplex PCRs; they successfully detected *nuc*, *mec*A, and *van*A/B genes as specific markers for *S. aureus*, methicillin resistance, and vancomycin resistance. Amplification products of the predicted sizes (192, 310, and 420 bp) were detectable (Figure 2A). To further confirm the specificity of the designed primers, the amplified PCR products were sequenced, and the resulting nucleotide sequences had 99.53% similarity to *S. aureus nuc* MZ816763.1, 99.24% similarity to *S. aureus mec*A EF600988.1, and 99.49% similarity to *S. aureus van*A MK214489.1. They were deposited in GenBank under accession numbers ON934435, ON934436, and ON934437, respectively. The mPCR was then checked using the three primer pairs; it successfully detected *S. aureus* (*nuc*) and methicillin resistance (*mec*A) with LODs of 10^7^ CFU/mL and vancomycin resistance (*van*A/B) with an LOD of 10^5^ CFU/mL (Figure 2B, Table 2).

### 3.2. The Developed mPCR Is Specific

Other microbial strains were tested to confirm the specificity of the developed mPCR. The assay successfully detected *S. aureus* ATCC 25923 with one single band of 192 bp, corresponding to the partial amplification of the *nuc* gene (Appendix A). The MRSA strain S43 produced two bands of 192 bp and 310 bp, corresponding to the partial amplification of the *nuc* and *mec*A genes; the vancomycin-resistant *E. faecalis* strain Y3 produced a single band of 420 bp, corresponding to the partial amplification of *van*A/B (Appendix A). No PCR amplification products were detected on agarose gel when testing *A. baumannii* ATCC 19606, *E. coli* ATCC 25922, *E. faecium* ATCC 27270, *E. faecalis* ATCC 19433, *K. pneumoniae* ATCC 10031, or *P. aeruginosa* ATCC 27856 (Appendix A). The developed mPCR specifically detected the three genes simultaneously in pooled DNA from S31 and E25 (Figure 3).

### 3.3. Detection of MRSA and Vancomycin Resistance Using the MBDLFA

The optimized mPCR was modified to develop the MBDLFA. The developed MBDLFA was specific and successfully detected *nuc*, *mec*A, and *van*A/B from the DNA extracted from pure bacterial colonies as well as from pooled DNA of S31 and E25, without nonspecific products under all tested conditions (Figure 4). No products were detectable when the developed MBDLFA was applied to other Gram-positive and Gram-negative pathogens and one single band corresponding to the *nuc* gene was detectable when the DNA extracted from *S. aureus* ATCC 25923 was tested (Appendix A).

### 3.4. Detection of MRSA and Vancomycin Resistance in Artificially Inoculated Blood Using the MBDLFA

The DNA was extracted from blood samples artificially inoculated with either S31 or E25 and the developed MBDLFA was performed on the extracted DNA. The assay successfully detected *S. aureus*, methicillin resistance, and vancomycin resistance in the DNA extracted directly from the artificially inoculated blood samples with LODs of 10^7^ CFU/mL, 10^8^ CFU/mL, and 10^4^ CFU/mL for *nuc, mec*A, and *van*A/B genes, respectively (Figure 5A,B, Table 2), using the designed C-PAS. When the products were visualized on 1.5% agarose gel stained with ethidium bromide, the LODs of the assay from artificially inoculated blood samples was 10^5^ CFU/mL for both the *nuc* and *mec*A genes and 10^4^ CFU/mL for the *van*A/B genes (Figure 5C, Table 2).

## 4. Discussion

SAB is accompanied by high morbidity and mortality rates [4] that are affected by several factors, including the antimicrobial resistance pattern and proper timing of antimicrobial administration [11,28]. The available bacteremia diagnosis methods depend on blood culturing, which is usually time-consuming and subject to possible failure [6,28].

In this study, *nuc, mecA,* and *van*A/B genes encoding the thermonuclease enzyme, penicillin-binding protein 2a (PBP2a), and D-Ala-D-Lac ligase were used as markers for *S. aureus* identification, methicillin resistance, and vancomycin resistance, respectively. The *nuc* gene is used as a specific marker for *S. aureus* identification using molecular techniques [21]. The PCR detection of *mec*A is the gold-standard technique for MRSA identification [22]. Several studies have designed mPCRs for identifying MRSA using primers specific for the *nuc* and *mec*A genes [29,30]. Other genes have also been used for PCR identification of *S. aureus*, such as 16S rRNA and *vic*K [31,32]. Sometimes, other target genes, such as *fem*B, are incorporated into the mPCR to enhance the reaction specificity and detectability of methicillin resistance [33]. *van*A is the common detectable ligase-encoding gene in *S. aureus* that is responsible for vancomycin resistance, whereas *van*B is rarely detected [23,34].

The designed primers were checked in uniplex PCR reactions. Single bands of the expected size were produced, and sequencing of the purified PCR products further confirmed their specificity. Combining the primers in a mPCR and testing different strains of MRSA and vancomycin-resistant enterococci resulted in the expected products without non-specific bands. The specificity of the developed mPCR was confirmed by testing other bacteremia-causing Gram-positive and Gram-negative pathogenic species. The use of mPCR is advantageous over the uniplex format in terms of time and money during diagnosis and screening reactions.

The developed mPCR reaction successfully detected MRSA and vancomycin resistance with LODs of 10^7^, 10^7^, and 10^5^ CFU/mL for the *nuc*, *mec*A, and *van* A/B genes, respectively, in the DNA extracted from the tested isolates. A lower LOD was reported in some studies where an mPCR was developed for identifying *S. aureus* and methicillin resistance using *nuc* and *mec*A [35,36]. However, our LOD (10^5^ CFU/PCR) is nearly similar to that reported in other studies (10^4^ to 10^6^ CFU/PCR) [37,38]. A previous mPCR developed by Okolie and colleagues [39] detected methicillin and vancomycin resistance as well as other *S. aureus* species-specific genes with an LOD of 10^4^ CFU/mL, which is slightly lower than our detected LOD.

Using a DNA-based lateral flow assay to detect PCR amplicons is an easy, fast, and user-friendly method for application as a point-of -care diagnostic [16], besides being safe due to the lack of use of the carcinogenic ethidium bromide. We successfully developed an MBDLFA capable of the specific detection of *S. aureus*, MRSA, and vancomycin resistance. The MBDLFA successfully detected *nuc*, *mec*A, and *van*A/B in the DNA extracted directly from artificially inoculated blood samples with LODs of 10^7^, 10^8^, and 10^4^ CFU/mL (equivalent to 10^4^, 10^5^, and 10 CFU/reaction), respectively. The lower LOD detected for the *van*A/B genes in all cases compared with the *nuc* and *mec*A genes may be attributed to the available genetic copies for the tested genes where *van*A/B are sometimes plasmid encoded. This LOD was higher than that for the same reactions visualized on an agarose gel stained with ethidium bromide (10^5^, 10^5^, and 10^4^ CFU/mL, equivalent to 10^2^, 10^2^, and 10 CFU/reaction for *nuc*, *mec*A, and *van*A/B, respectively). A nested uniplex PCR developed by Banada and colleagues [40] detected *S. aureus* directly from blood samples with LODs of 10 CFU/mL and 50 CFU/mL for *sod*A and *nuc* genes, respectively. The higher LOD in our reaction may be due to the multiplex nature of the reaction, where the sensitivity of PCR decreases as the number of amplified genes increases, and the use of the nested PCR technique in Banada et al. study. The enhanced sensitivity of the nested PCR arises from the increased total number of cycles used in the two reaction rounds [41]. Other studies designed real-time PCR reactions to detect *S. aureus* and/or methicillin resistance directly from blood samples with a lower detection limit of 10^3^ CFU/mL [42,43]. The real-time PCR technique is not available in all laboratories, especially in developing countries, in addition to its high cost and the need for more trained personnel compared with conventional PCR. Several PCR-based lateral flow assays have been developed to detect different pathogens, resistance genes, and microbial toxins [18,20,44,45,46]. A previous study reported the use of a multiple cross displacement amplification reaction coupled with a lateral flow assay to detect *S. aureus* and identify MRSA with an LOD as low as 10^3^ CFU/mL. However, multiple cross displacement amplification reactions use a large set of primers per reaction and suffer from false positive results due to contamination [47], which is not the case with our developed MBDLFA. Another developed PCR-based strip detected MRSA from positive blood cultures using a DNA lateral flow technique [45]. However, blood culture bottles are positive only when the microbial count exceeds 10^7^–10^8^ CFU/mL [48].

The developed MBDLFA allows for the simultaneous detection of *S. aureus*, MRSA, and vancomycin resistance directly from blood samples and/or cultures in two hours. Rapid diagnosis is a critical requirement for the proper treatment of SAB [7]. Bacteremia diagnosis depends mainly on the blood culture results; blood cultures require at least 5 h to become positive and may need 15 h depending on the initial bacterial load and antibiotic treatment at the sampling time [43,49,50]. After confirmation of positive blood cultures, antimicrobial testing is performed. The results of identification and antimicrobial susceptibility are only available after 72 h [50]. During that time, empirical treatment is usually administered, which could involve the unnecessary use of vancomycin as an anti-MRSA agent [10]. This may alter the patient’s microbiome leading to opportunistic infections, subject the patient to drug-related toxicity, and increase the cost of hospitalization. In addition, improper antibiotic use can select for resistant strains [28,51]. Using the developed MBDLFA on DNA extracted from positive blood cultures will also save the time required for traditional testing of MRSA and vancomycin resistance.

The developed MBDLFA permits the use of as little as 100 µL of blood for DNA extraction; a higher blood volume may be used to enhance the LOD of the assay. Other blood culture techniques that require higher blood volumes allow for higher sensitivity [42] but can be problematic when it is difficult to obtain large blood volumes, as in pediatric cases [28].

Many studies regarding molecular-based methods for pathogen identification and antimicrobial resistance marker detection depend on positive blood cultures [52]. Before culturing, the initial bacterial count in blood is less than 1 CFU/mL in half of the patients with bacteremia [53]. The minimum number of bacteria present in blood culture before a culture is positive is 10^8^ CFU/mL [48], which is similar or slightly higher than the detection limit of the developed MBDLFA. After a positive blood culture, the suggested tests are either in situ hybridization-based methods, DNA-microarray-based hybridization technologies, nucleic-acid-amplification-based methods, or combined platforms [52]. Although in situ hybridization and DNA-microarray based methods are as fast as PCR-based techniques, they have a higher or similar detection limit (10^6^ CFU/mL). These methods imply the need for expensive instrumentation and highly trained personnel, which may not be available in developing countries as a point-of-care test. In contrast to the developed MBDLFA, the in situ hybridization method does not provide information about the antimicrobial susceptibility [6,52]. It is worth mentioning that about 50% of bacteremia cases occur with a negative blood culture result due to antibiotic use or a low blood bacterial load, which will delay the introduction of proper treatment [28]. Molecular methods performed directly on blood allow the detection of bacteremia even if the blood culture is negative, especially after the start of antibiotic treatment [42,54].

Some PCR-based systems are commercially available but are intended mainly for identification purposes; some tests allow for MRSA detection directly from blood [6]. These blood-based MRSA detection tests depend mainly on real-time PCR or conventional PCR followed by sequencing, which requires expensive kits and instrumentation and trained personnel, which is not readily available in lower-resourced places.

All PCR-based assays suffer false positive results due to contaminating DNA from the used reagents or environment, DNA from dead bacteria (DNAemia), or infections already controlled by the immune system. Quantitative real-time PCR helps better interpret positive results, as the low abundance of bacteria usually characterizes contamination [28]. It is always a good practice to make parallel use of blood culture as the gold-standard method for diagnosis of bacteremia with PCR-based assays [51]. This will allow for the rapid detection of bacteremia with a high blood bacterial load and commencement of proper treatment within the first 2 h of sampling. The blood culture results can be used as a confirmatory tool. Using PCR-based methods for MRSA identification in bacteremia patients resulted in a timely shift to effective therapy, with shorter hospital stays and lower hospitalization costs [55].

## 5. Conclusions

The developed MBDLFA represents a rapid and cheap point-of-care diagnostic for simultaneous MRSA and vancomycin detection in bacteremia patients at the admission site and, consequently, prompt and proper treatment. This will reduce the morbidity and mortality rates associated with SAB and the hospitalization time and cost.

## Figures and Tables

**Figure 1 diagnostics-12-02691-f001:**
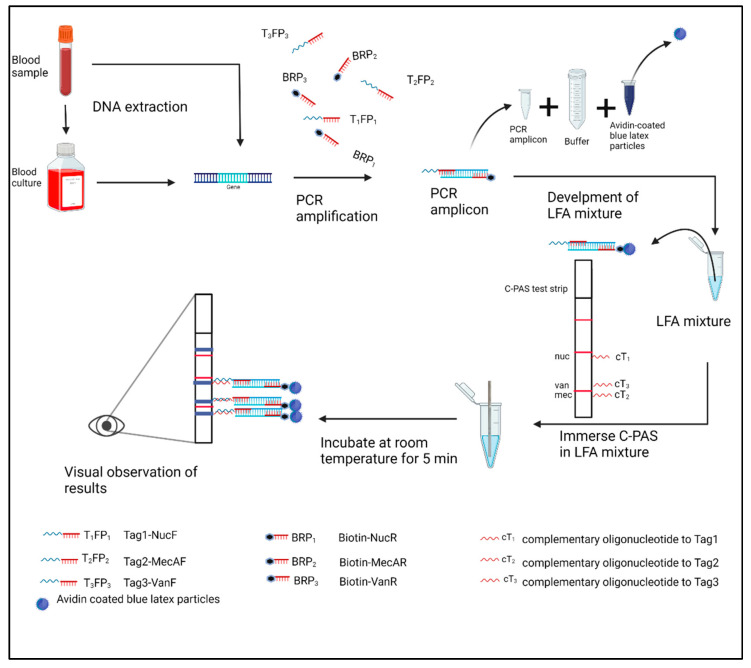
Schematic diagram of the of the multiplex polymerase chain reaction (PCR)-based DNA lateral flow assay (MBDLFA) workflow; created with biorender.com. The designed chromatography-printed array strip (C-PAS) containing complementary oligonucleotides to each single-stranded tag and indicating the expected positions of the generated colored line specific to each tag is shown. The red lines in strip represent position markers that help localization of test lines. Tag1-NucF, Tag2-MecAF, Tag3-VanF: single-stranded oligonucleotides-tagged forward primers for amplification of *nuc*, *mec*A, and *van* genes, respectively; Biotin-NucR, Biotin-MecAR, Biotin-VanR: biotin-labelled reverse primers for amplification of *nuc*, *mec*A, and *van* genes, respectively.

**Figure 2 diagnostics-12-02691-f002:**
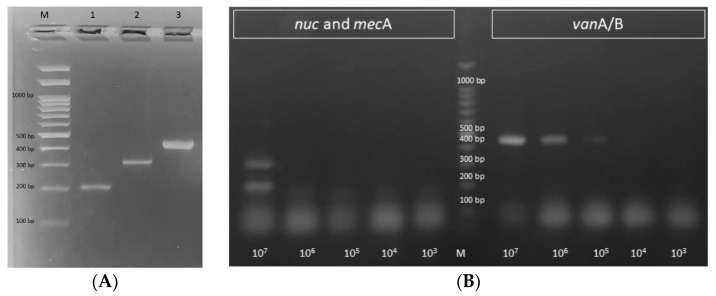
Agarose gel visualization of the amplification products of the developed: (**A**) uniplex PCR, where: M is a 100 bp DNA ladder, lanes 1–3 show the amplification products of *nuc*, *mec*A, and *van*A/B genes with the expected product sizes of 192, 310, and 420 bp, respectively, and (**B**) mPCR performed on the genomic DNA extracted from different dilutions of S31 and E25 (from 10^7^ to 10^3^ CFU/mL). It successfully detected *S. aureus* (*nuc*; 192 bp) and methicillin resistance (*mec*A; 310 bp) with a LODs of 10^7^ CFU/mL and vancomycin resistance (*van*A/B; 420 bp) with an LOD of 10^5^ CFU/mL. M: 100 bp DNA ladder.

**Figure 3 diagnostics-12-02691-f003:**
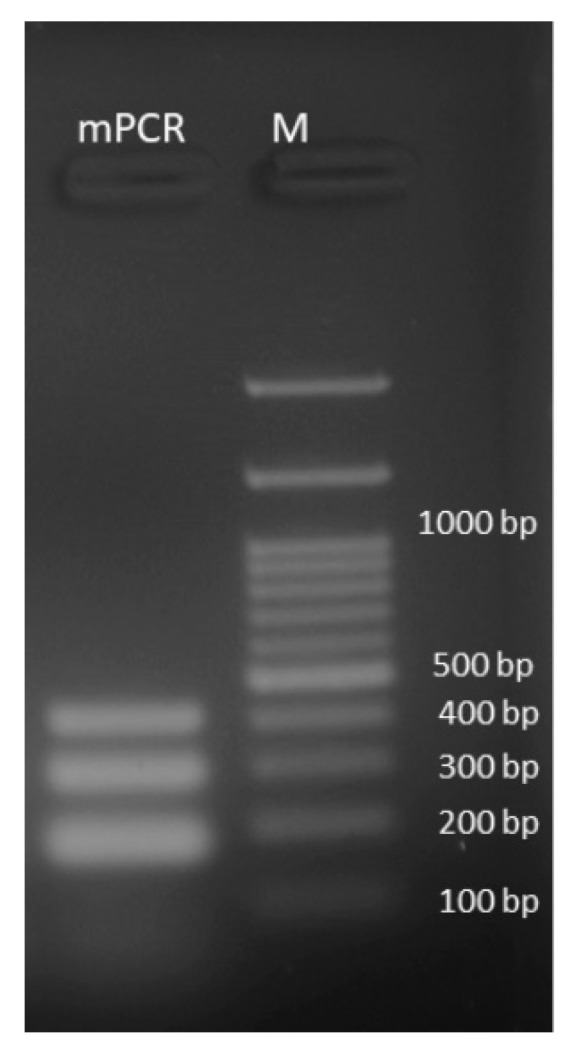
Agarose gel visualization of the amplification products of the developed mPCR performed on the pooled genomic DNA of S31 and E25. The developed mPCR specifically detected the three genes simultaneously, (*nuc*: 192 bp; *mec*A: 310 bp, and *van*A/B: 420 bp). M: 100 bp DNA ladder.

**Figure 4 diagnostics-12-02691-f004:**
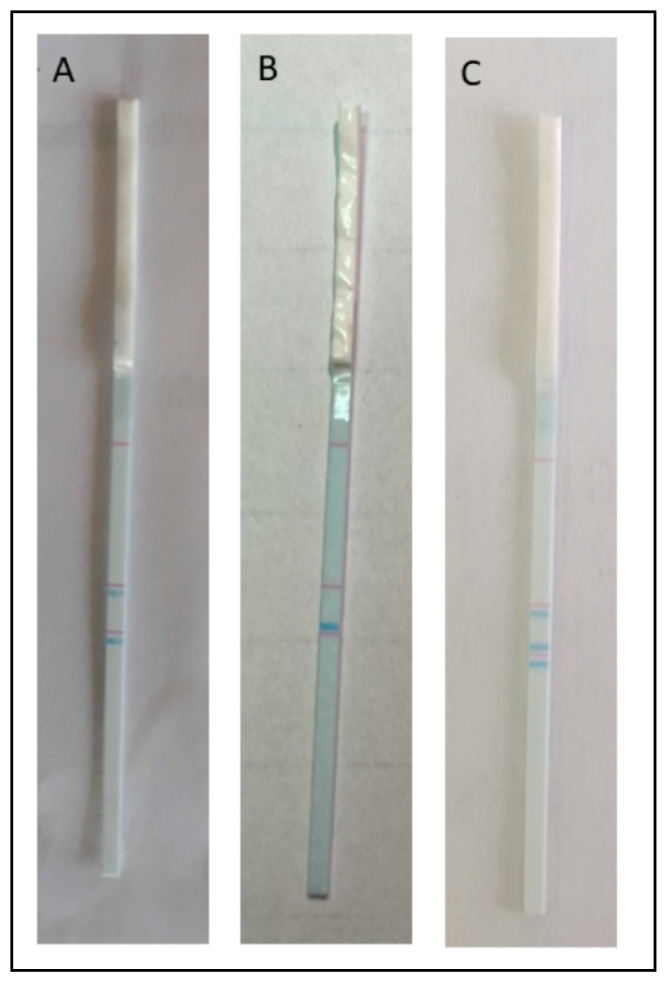
The developed multiplex polymerase reaction (mPCR)-based DNA lateral flow assay (MBDLFA) was specific and successfully detected: (**A**) *nuc* and *mec*A from genomic DNA extracted from *Staphylococcus aureus* S31; (**B**) *van*A/B from genomic DNA extracted from *Enterococcus faecalis* E25; and (**C**) *nuc*, *mec*A, and *van*A/B from pooled genomic DNA of S31 and E25, using the designed chromatography-printed assay strip (C-PAS). The red lines on the strip represent position markers that help localization of the test lines.

**Figure 5 diagnostics-12-02691-f005:**
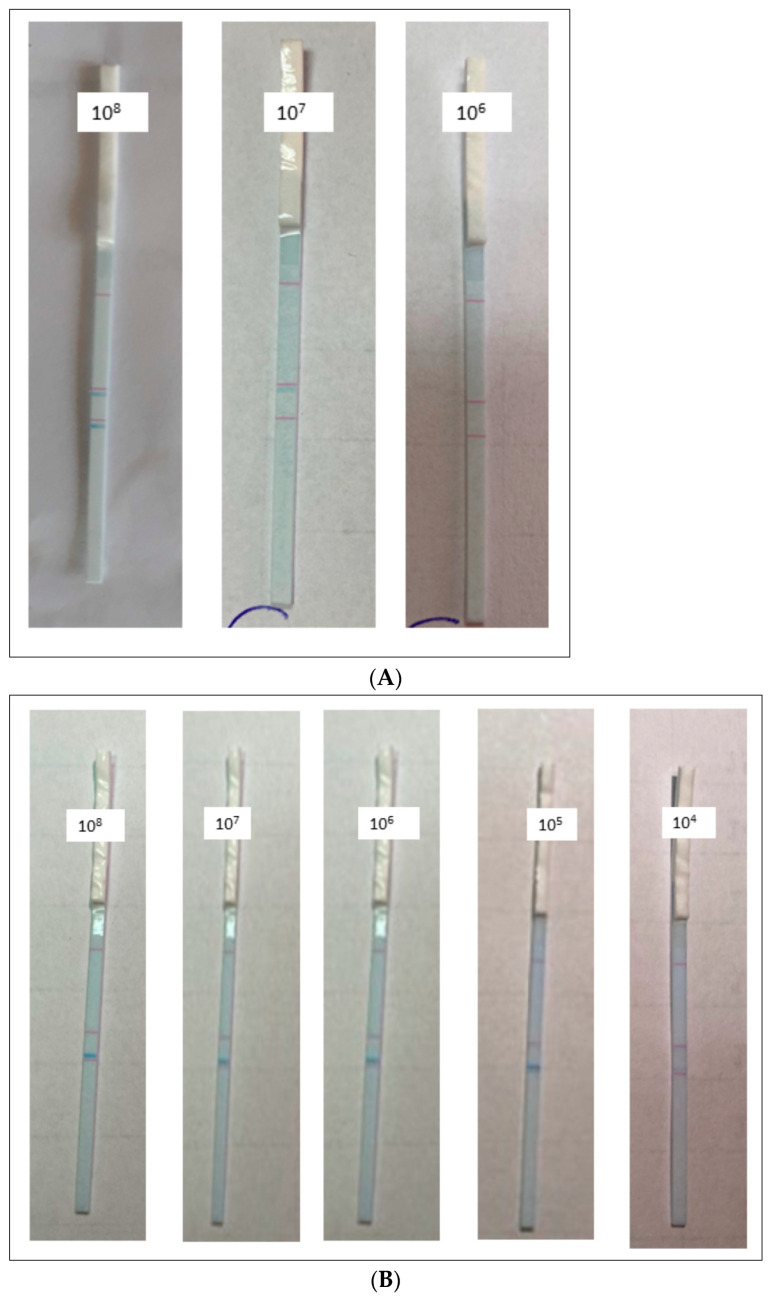
The limit of detection (LOD) of the developed multiplex polymerase reaction (mPCR)-based DNA lateral flow assay (MBDLFA) using the DNA extracted from blood samples artificially inoculated with either *Staphylococcus aureus* S31 or *Enterococcus faecalis* E25. The assay successfully detected: (**A**) *S. aureus* (*nuc*) and methicillin resistance (*mec*A) in S31 with LODs of 10^7^ CFU/mL and 10^8^ CFU/mL, respectively, using the designed chromatography-printed assay strip (C-PAS), and (**B**) vancomycin resistance (*van*A/B) in E25 with an LOD of 10^4^ CFU/mL, using the designed C-PAS. (**C**) The products were visualized on a 1.5% agarose gel stained with ethidium bromide, the LODs of the assay from artificially inoculated blood samples was 10^5^ CFU/mL for both the *nuc* and *mec*A genes and 10^4^ CFU/mL for the *van*A/B genes. The red lines on the strips in (**A**,**B**) represent position markers that help the localization of test lines.

**Table 1 diagnostics-12-02691-t001:** Primers used in the study and the PCR amplicon size.

Primer Name	Target Gene	Sequence (5′-3′)	Product Size	Source
^1^ NucF	*nuc*	GCGATTGATGGTGATACGGTT	192	[24]
^1^ NucR		TGACCTTTGTCAAACTCGACTTC		This study
^1^ MecAF	*mec*A	GTAGAAATGACTGAACGTCCGATA	310	[25]
^1^ MecAR		CCAATTCCACATTGTTTCGGTCTA		[25]
^1^ VanF	*van*A/B	ATCGTTGACATACATCGTTGCG	421	This study
^1^ VanR		CTGTATCCGTCCTCGCTCCT		This study
^2^ Tag1-NucF	*nuc*	[Tag1]-spacer-GCGATTGATGGTGATACGGTT	192	This study
^2^ Biotin-NucR		[Biotin]-TGACCTTTGTCAAACTCGACTTC		This study
^2^ Tag2-MecAF	*mec*A	[Tag2]-spacer-GTAGAAATGACTGAACGTCCGATA	310	This study
^2^ Biotin-MecAR		[Biotin]-CCAATTCCACATTGTTTCGGTCTA		This study
^2^ Tag3-VanF	*van*A/B	[Tag3]-spacer-ATCGTTGACATACATCGTTGCG	421	This study
^2^ Biotin-VanR		[Biotin]-CTGTATCCGTCCTCGCTCCT		This study

^1^ Primers used for conventional uniplex and multiplex polymerase chain reactions. ^2^ Primers used in multiplex polymerase chain reaction-based DNA lateral flow assay.

**Table 2 diagnostics-12-02691-t002:** Limit of detection, time for detection, and specificity of the tested methods.

Tested Method	Limit of Detection (CFU/mL)			Time for Detection	Specificity
	*nuc*	*mec*A	*van*A/B		
Conventional mPCR followed by gel electrophoresis	10^7^	10^7^	10^5^	3 h * or more depending on the number of samples.	Specific
MBDLFA on DNA extracted from artificially spiked blood followed by gel electrophoresis	10^5^	10^5^	10^4^	3 h * or more depending on the number of samples	Specific
MBDLFA on DNA extracted from artificially spiked blood followed by using C-PAS	10^7^	10^8^	10^4^	2 h *	Specific

mPCR: multiplex polymerase chain reaction; MBDLFA: mPCR-based DNA lateral flow assay; C-PAS: chromatography-printed assay strip. * This is dependent on the time needed for the blood culture to be positive if needed.

## Data Availability

The data presented in this study are available in the article text and Appendix A.

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
