# Peer review of "Development of a Multiplex Polymerase Chain Reaction-Based DNA Lateral Flow Assay as a Point-of-Care Diagnostic for Fast and Simultaneous Detection of MRSA and Vancomycin Resistance in Bacteremia"

_diagnostics, 2022, doi:10.3390/diagnostics12112691_

Round 1
Reviewer 1 Report
Comments for “Development of a multiplex polymerase chain reaction-based DNA lateral flow assay as a point-of-care diagnostic for fast and simultaneous detection of MRSA and vancomycin resistance in bacteremia” (diagnostics-1959524)
In this paper, the authors developed a lateral flow assay for simultaneous detection of nuc, mecA, and vanA/B. The topic is interesting and meaningful. However, there are some key concerns from this reviewer, which need to be addressed seriously.
1) The relationship between the mPCR and LFAs is not clear. Were the LFAs used to detect amplicons after mPCR? Or directly detecting the genes from sample?
2) The working principle of mPCR and LFAs is not clear. For example, how to capture double-strand DNA on the T-line of the LFA after PCR?
3) The signal on LFAs was not readable.
4) All the figures are in low quality, which have to be improved before publishing.
5) English grammar and language check throughout the manuscript are also necessary.
Author Response
The authors thank the reviewer for useful comments and suggestions. All the reviewer's concerns were addressed, as stated below.
1) The relationship between the mPCR and LFAs is not clear. Were the LFAs used to detect amplicons after mPCR? Or directly detecting the genes from sample?
The LFA was used to detect amplicons after mPCR and not genes from the sample. The general principle of LFA has been described in introduction (Line 93-105) and stated in detail in the methods section (Lines 197-201, 210-217). An illustration of the workflow of the MBDLFA has been added as Figure 1.
2) The working principle of mPCR and LFAs is not clear. For example, how to capture double-strand DNA on the T-line of the LFA after PCR?
The principle of the developed MBDLFA has been stated in detail. DNA lateral flow assays, depending on single-tag hybridization technology, are developed as dipstick strips for the detection of PCR amplicons. Single-tag hybridization depends on using biotinylated oligonucleotides and a single-stranded tag-spacer sequence for preparing the PCR amplicons that bind streptavidin-coated blue latex particles. The product is then hybridized to a complementary probe immobilized on the membrane strip, producing a colored line. The membrane test strip is a nitrocellulose paper with different pads that allows capillary sample flow. The sample is applied onto a sample pad and moves by capillary flow to reach the conjugate pad, where it reacts with the complementary probe, thus allowing product visualization. The principle has been further explained in introduction (Lines 93-105) and methods sections (Lines 197-201, 210-217) and illustrated in Figure 1.
3) The signal on LFAs was not readable.
The authors thank the reviewer for his note and the quality of all figures has now been enhanced based on journal recommendations.
4) All the figures are in low quality, which have to be improved before publishing.
The quality of all figures has now been enhanced based on journal recommendations.
5) English grammar and language check throughout the manuscript are also necessary.
The manuscript has been checked by an English speaker.
Reviewer 2 Report
This article reported the development of a multiplex PCR-based DNA lateral flow assay for MRSA and vancomycin resistance in bacteremia detection. The author demonstrates the use of lateral flow assay to detect PCR amplification product which is specific to the gene nuc, mecA, and vanA/B. The author showed the validation of designed primers and analytical performance on lateral flow assay of the proposed method. This manuscript is well-organized and technically sound. However, some issues may be raised that need to be considered before further consideration.
1. For Table 1. I would like to get clarification on the designed primer table. As the author mentioned in the last column "source" and determined that “This study”. Does it mean the author has 100% designed these primers? As I quickly checked the sequence shown in this table has been used elsewhere. If this sequence has been studied before, please have proper citations.
2. In section 2.6, what does it mean for C-PAS strip? as it first used here and has never been described before. Moreover, the information about MBDLFA is missing. There is no detail on lateral flow assay preparation. As a result, I am also confused with the result in Figure 4 and Figure 5A, 5B since I saw the red line there, but there is no information mentioned about using gold nanoparticles (if used to illustrate the red line) as well as how the lateral flow assay can be controlled. Please provide more information.
3. According to insufficient lateral flow assay information, it also raises another question about analytical performance. How does the author evaluate the limit of detection in lateral flow assay? If the author uses naked-eye evaluation, how could the author confirm that it is free of bias? as the lateral flow assay gives us a yes/no answer (qualitative result). Furthermore, the author mentioned the false positive issue due to the larger primers used in this study. Has the author altered the order of detection lines? Is the lateral flow assay still providing the same result? Please add more information and discuss.
4. I understand that the combination of lateral flow assay and PCR amplicons may replace some post-PCR steps such as gel electrophoresis, pyrosequencing, or fluorescent-probe hybridization. However, is there any comparative method elsewhere? Please include in this paper, compare the efficiency with those, and discuss why lateral flow assay is better than other methods.
Author Response
The authors thank the reviewer for useful comments and suggestions. All the reviewer's concerns were addressed, as stated below.
- For Table 1. I would like to get clarification on the designed primer table. As the author mentioned in the last column "source" and determined that “This study”. Does it mean the author has 100% designed these primers? As I quickly checked the sequence shown in this table has been used elsewhere. If this sequence has been studied before, please have proper citations.
The authors thank the reviewer for his comment. Actually the primers were 100% designed in this study but when we searched it again based on the reviewer's suggestion we found that 3 of them were previously used in other studies so we corrected this error and included the correct citation.
- In section 2.6, what does it mean for C-PAS strip? as it first used here and has never been described before. Moreover, the information about MBDLFA is missing. There is no detail on lateral flow assay preparation. As a result, I am also confused with the result in Figure 4 and Figure 5A, 5B since I saw the red line there, but there is no information mentioned about using gold nanoparticles (if used to illustrate the red line) as well as how the lateral flow assay can be controlled. Please provide more information.
C-PAS means chromatography-printed array strip, and this has been added in the text (Line 200) and explained in the relevant figure captions. The details about LFA preparation have been stated in details in the methods section (Lines 210-217), where the mPCR products for MBDLFA (10 µL) were mixed with 10 µL dilution Buffer, and 1 µL of latex solution containing avidin-coated blue beads. The designed C-PAS test strip was immersed in the mixture and the results were visualized, after a 5 min incubation period at room temperature, by blue-colored lines corresponding to each tested gene (Figure 1). An illustration of the workflow of the MBDLFA has been added in Figure 1. The red lines in the strip represent a position marker that helps the localization of test lines. This has been stated in relevant figure captions. The developed MBDLFA was controlled by testing the DNA extracted from different Gram-positive and Gram-negative pathogens and the pooled DNA from S31 and E25 (Lines 217-220). The developed MBDLFA successfully detected nuc, mecA, and vanA/B from the DNA extracted from pure bacterial colonies as well as from pooled DNA of S31 and E25, without non specific products under all tested conditions (Figure 4). No products were detectable when the developed MBDLFA was applied to other Gram-positive and Gram-negative pathogens (Lines 293-298).
- According to insufficient lateral flow assay information, it also raises another question about analytical performance. How does the author evaluate the limit of detection in lateral flow assay? If the author uses naked-eye evaluation, how could the author confirm that it is free of bias? as the lateral flow assay gives us a yes/no answer (qualitative result). Furthermore, the author mentioned the false positive issue due to the larger primers used in this study. Has the author altered the order of detection lines? Is the lateral flow assay still providing the same result? Please add more information and discuss.
The results of the developed MBDLFA were visualized by the naked eye of two independent technicians to avoid any bias during result interpretation. This has been stated in text (Lines 243-245). The false positive results due to large number of primer sets was reported in a previous article using multiple cross-displacement amplification reactions which don’t occur in case of our developed mPCR (Lines 399-402). Altering the detection line alignment won’t change the results as each detection line has immobilized oligonucleotides complementary to that on the 5’ terminus of the forward primer used in amplicon production, which allows binding the specified amplicon only if present. The details about LFA preparation have been described in introduction (Lines 93-105) and methods sections (Lines 197-201, 210-217), and an illustration of the workflow of the MBDLFA has been added as Figure 1.
- I understand that the combination of lateral flow assay and PCR amplicons may replace some post-PCR steps such as gel electrophoresis, pyrosequencing, or fluorescent-probe hybridization. However, is there any comparative method elsewhere? Please include in this paper, compare the efficiency with those, and discuss why lateral flow assay is better than other methods.
LFA is used as a mean of detection of amplicons generated by mPCR. It can replace post-PCR steps as gel electrophoresis. The LFA is an easy, fast, and user-friendly method for application as a point-of -care diagnostic, besides being safe due to the lack of use of the carcinogenic ethidium bromide (Line373-375). Comparison to other molecular methods for bacteremia diagnosis was made. Other blood-based MRSA detection tests depend mainly on real-time PCR or conventional PCR followed by sequencing, which require expensive kits and instrumentations and trained personnel, which are not readily available in low-resourced places (Lines 385-395). Other molecular-based systems for microbial identification and antimicrobial susceptibility detection during diagnosis of bloodstream infection are also available, such as in-situ hybridization and DNA-microarray-based methods. Although these methods are as fast as PCR-based techniques; however, they have a higher or similar detection limit (106 CFU/mL). These methods imply the need for expensive instrumentation and highly trained personnel, which are not available for developing countries as a point-of-care test. In contrast to the developed MBDLFA, in situ hybridization method doesn’t provide information on the antimicrobial susceptibility. This has been included in discussion (Lines 440-448).
Reviewer 3 Report
The author(s) developed a multiplex polymerase reaction (mPCR)-based DNA lateral flow assay (MBDLFA) for detecting S. aureus, MRSA, and vancomycin resistance directly from blood and blood cultures. The sensitivity, specificity, and LOD of the developed reaction were also determined and further optimized. This work is well-organized with a smooth logical line which can give readers a better understanding of the contents. However, the novelty of the work raises the biggest concern. The following are some concerns about this work before it can be accepted.
1. Could the author(s) clarify what’s the needed initial concentration of the target sequences for an accurate detection? It seems that the competition of primers for the target sequences and the reagents will significantly affect the detection results.
2. To make it clearer, the author(s) should show experiment results for the optimization of the most adequate concentrations of the target sequence, proving that all sequences will be amplified similarly and thus eliminating nonspecific amplification products.
3. Could the author(s) explain more about the comparison between multiplex PCR and singleplex PCR (amplification of a single target sequence of one microorganism) for MRSA and vancomycin resistance detection to prove its superiority and necessity? I suggest the author(s) add a table showing the comparison results, including detection sensitivity, detection time, detection specificity, initial concentration, etc.
Author Response
The authors thank the reviewer for useful comments and suggestions. All the reviewer's concerns were addressed, as stated below.
- Could the author(s) clarify what’s the needed initial concentration of the target sequences for an accurate detection? It seems that the competition of primers for the target sequences and the reagents will significantly affect the detection results.
The minimum needed initial concentration of target sequences was determined through the limit of detection determination (Line 178-186) and was stated in Table 2 for different tested techniques.
- To make it clearer, the author(s) should show experiment results for the optimization of the most adequate concentrations of the target sequence, proving that all sequences will be amplified similarly and thus eliminating nonspecific amplification products.
The minimum needed initial concentration of target sequences (most adequate conc) was determined through limit of detection determination and the procedure is described in methods section (Lines 178-186). Nonspecific products were not detectable under any tested concentration as stated in Figure 4 and Figure 5. The optimization of the reactions developed involved testing different primers and Mg2+ concentrations, a gradient of annealing temperatures ranging from 55-60°C, and high and low concentrations of DNA (Line 203-205) to determine the optimum reaction components and conditions (Line 205-210).
- Could the author(s) explain more about the comparison between multiplex PCR and singleplex PCR (amplification of a single target sequence of one microorganism) for MRSA and vancomycin resistance detection to prove its superiority and necessity? I suggest the author(s) add a table showing the comparison results, including detection sensitivity, detection time, detection specificity, initial concentration, etc.
The authors thank the reviewer for his comment and suggestion. The use of mPCR is advantageous over uniplex type in terms of saving time and money during diagnosis and screening reactions. This has been indicated in the discussion (Lines 361-363). Based on the proved specificity of the developed primers by uniplex format (Line 356-358), the mPCR format was developed and used for further testing. All comparison results of detection, limit, detection time and specificity were included in Table 2.
Round 2
Reviewer 2 Report
Thank you for your clarification. The author has addressed and modified the manuscript as suggested. However, I concern about MBDLFA figures. It would be good for readers to better understand the comparison between MBDLFA if the author could improve the quality of the figure. My suggestion would be taking the figure in the light box to avoid shadow, so the reader can better perceive the blue line clearer. Then, I would like to recommend for the publication.
Author Response
We would like to thank the reviewer for his comments. We have removed the option of compressing figures in word files to get pictures with better quality. Figures 5A and 5B were replaced by better ones. The uploaded figures folder contains individual figures with good quality. The blue lines are now perceived better. Unfortunately, a light box is unavailable in our lab.